# Potentially Avoidable Hospitalizations by Asthma and COPD in Switzerland from 1998 to 2018: A Cross-Sectional Study

**DOI:** 10.3390/healthcare11091229

**Published:** 2023-04-26

**Authors:** Alexandre Gouveia, Charlène Mauron, Pedro Marques-Vidal

**Affiliations:** 1Department of Ambulatory Care, Center for Primary Care and Public Health (Unisanté), University of Lausanne, Rue du Bugnon 44, 1011 Lausanne, Switzerland; 2Department of Vulnerabilities and Social Medicine, Center for Primary Care and Public Health (Unisanté), University of Lausanne, Rue du Bugnon 44, 1011 Lausanne, Switzerland; 3Division of Internal Medicine, Lausanne University Hospital (CHUV), Rue du Bugnon 46, 1011 Lausanne, Switzerland

**Keywords:** asthma, chronic obstructive pulmonary disease, potentially avoidable hospitalizations, quality of care, primary care, public health

## Abstract

Potentially avoidable hospitalizations (PAH) are commonly used as an indicator for healthcare quality and primary care performance. However, data are usually presented in a restricted timeframe and for a specific region, limiting the identification of trends and national patterns. We aimed in this study to calculate rates, identify clinical determinants, and estimate costs of PAH for two frequent lung diseases (asthma and COPD) in Switzerland between 1998 and 2018 using hospital discharge data available for patients aged ≥20 years. PAH were defined according to the Health Care Quality Indicators Project (HCQIP) from the Organisation for Economic Co-operation and Development (OECD). The distribution of PAH in seven administrative regions (Leman, Mittelland, Northwest, Zurich, Eastern, Central, and Ticino) was calculated, along with PAH-associated total hospital days and Diagnosis-Related Group (DRG) estimated costs. Totals of 25,260 PAH for asthma and 135,069 PAH for COPD were identified in the 20-year period. The standardized rates of PAH per 100,000 people for asthma fluctuated from 18.7 in 1998 to 22.5 on 2018. The standardized rates of PAH per 100,000 people from COPD almost doubled from 77.4 in 1998 to 142.7 in 2018. In 2018, the estimated total costs of PAH amounted to 7.7 million CHF for asthma and 91.2 million CHF for COPD. We conclude that PAH for asthma and COPD represent a significant and unnecessary burden and costs of hospitalizations in Switzerland.

## 1. Introduction

Asthma is a chronic inflammatory respiratory disease affecting from 1 to 8% of the population [1]. Asthmatic patients can be asymptomatic during a prolonged period, but they can also experience acute and severe exacerbations that force them to seek urgent healthcare. A certain number of asthmatic patients are admitted to hospital each year, eventually because of suboptimal disease management in the ambulatory care setting [2].

Chronic obstructive pulmonary disease (COPD) was the third leading cause of mortality worldwide in 2019 according to the World Health Organization (WHO) [3]. COPD is a clinical respiratory syndrome causing persistent and progressive symptoms, such as difficulty in breathing, associated with cough and/or sputum [4]. The progressive course of the disease and the occurrence of exacerbations can be avoided by adequate follow-up and therapeutic optimization in the primary care setting, therefore reducing the burden of COPD emergency and hospital admissions [5,6].

Switzerland is one of the richest countries in the world and has a high performing healthcare system, although rather complex and expensive [7]. In 1997, a total of 406 public hospitals and private clinics existed in Switzerland, but, in recent decades, the hospital network was significantly modified, mostly due to restructuring and fusion of hospitals. In 2018, even if only 281 private and public hospitals were functioning, the hospital network is considered as adequate and well-functioning [8]. Primary care in Switzerland is based on fee-for-service and provided by independent practitioners that work in private practices. Primary care physicians are regularly involved in the management of chronic diseases, such as asthma and COPD. However, from 1992 to 2012, the Swiss primary care workforce became older, more female, and worked fewer hours per week. Thus, it is likely that primary care accessibility has diminished during that period as a consequence of the workforce evolution [9].

Potentially avoidable hospitalizations (PAH) can be used as a measurement of primary care accessibility in an established area [10]. Rates, trends, determinants, and costs of PAH are important factors in assessing the quality of healthcare and provide relevant information for healthcare resources planification in a specific area to optimize prevention and management of chronic diseases in the primary care setting. Yet, PAH seem to be associated with other factors than primary care accessibility or performance, such as socioeconomic attributes of the population and organization of the healthcare system [11]. This challenges interpretation of PAH as a healthcare quality indicator and makes them an interesting topic of research.

The Health Care Quality Indicator Project (HCQIP) from the Organisation for Economic Co-operation and Development (OECD) defined a set of indicators that has been used for healthcare systems monitoring and research. This common set of indicators allows a better interpretation of data and facilitates comparison between countries [12,13]. In the HCQIP, PAH for chronic diseases currently managed in the primary care setting (such as asthma, COPD, heart failure, or diabetes) are applied as indicators for assessing the quality and accessibility of primary care [14].

This study aimed to identify rates, determinants, hospital days, and costs of PAH related to lung diseases (asthma and COPD) among adult patients older than 20 years old between 1998 and 2018 in Switzerland. Few studies have assessed the rates, trends, and determinants of PAH in Switzerland, and, to the best of our knowledge, none has been recently conducted at a national level in such a wide timeframe [15,16,17].

## 2. Materials and Methods

### 2.1. Study Design and Length

We conducted a cross-sectional study using nationally representative data for Switzerland obtained from the Swiss Federal Office for Statistics for the period 1998 to 2018.

### 2.2. Variables, Data Sources, and Measurements

Hospital discharge data from 1998 to 2018 were formally requested to the Swiss Federal Office of Statistics (www.bfs.admin.ch) (accessed on 16 March 2023) to conduct this specific study. The data cover 98% of public and private hospitals within Switzerland and include all stays for each hospital, even those that last less than 24 h. The main cause for hospitalization and the comorbidities were coded using the 10th revision of the International Classification of Diseases (ICD-10) of the World Health Organization (WHO) [18]. The procedures were coded using the Swiss classification of surgical interventions (CHOP) [19]. Aside from the ICD-10 and CHOP codes, the data also contain information regarding gender, age (categorized into 5-year groups), administrative regions (the 26 Swiss cantons (federal states) were attributed to seven different regions, i.e., Leman, Mittelland, Northwest, Zurich, Eastern, Central, and Ticino; further information available as Appendix A), date of admission (limited to month and year), decision of admission (i.e., patient’s or doctor’s initiative, others), type of admission (planned or emergency), type of room (infirmary, semi-private, or private), and PAH-related total hospital days. For calculating the number of hospital days for each PAH, the length of stay (LOS) was expressed using the OECD definition [20].

As performed in another Swiss study conducted by Halfon et al., the Deyo adaptation of the Charlson index was used to assess comorbidities [21,22,23]. The index was computed using data from the current hospitalization, and patients were categorized into 0–1, 2–3, and 4+ score values.

### 2.3. Potentially Avoidable Hospitalizations

The definition for PAH for asthma and COPD was obtained according to the international OECD Health Care Quality Indicators Project criteria [14] (available as Appendix A).

### 2.4. Consequences of Potentially Avoidable Hospitalizations

The total number of days due to PAH was computed for each year. This number was then divided by 365 to obtain the number of beds that would be theoretically solely dedicated to PAH during that year.

Costs were computed using the Swiss Diagnosis-Related Group (DRG) system. As the value of the DRG point varies according to canton and it was not possible to obtain the values for each year from each canton, costs were computed by applying the cantons’ DRG value for year 2018 (Appendix A). Further, it was not possible to obtain the value for two small cantons (Ticino and Thurgau). Hence, calculations were limited to PAH data from year 2018 using DRG values for 2019 or 2020 from the twenty-four cantons. Data from the Vaud canton suggest that the value changed little (if any change occurred) with time: 10,650 CHF from 2016 to 2020 (1 CHF = 1.01 EUR or 1.07 USD as of 16 March 2023).

### 2.5. Inclusion and Exclusion Criteria

Only data related to adults (i.e., being at least in age group 20–24) were eligible for analysis. Patients coming from outside of Switzerland or hospitalizations with missing covariates were excluded.

### 2.6. Statistical Analysis

Analysis was conducted using Stata version 16.1 for Windows^®^ (Stata Corp, College Station, TX, USA). Descriptive results were expressed as number of hospitalizations (percentage) or as average ± standard deviation (SD). Standardized rates of PAH for each calendar year were computed via direct standardization using the 2013 standard EU population.

### 2.7. Ethics Statement

The data of the Swiss Federal Office of Statistics are available for research purposes, and, therefore, neither specific individual consent nor authorization from an Ethics Committee were needed. Data that are collected routinely in the Medical Statistics of Hospitals are de-identified. In Switzerland, an Ethics Commission approval is not needed for a study using anonymized data where patient identification is not possible [24]. According to this legal framework, this study was not submitted to an Ethics Commission evaluation and approval.

## 3. Results

### 3.1. Characteristics of Potentially Avoidable Hospitalizations

A total of 21,369,966 hospital admissions were collected in the initial database. Applying the PAH OECD criteria, an initial sample of 43,440 PAH for asthma was obtained, of which 14,094 (32.4%) were patients coming from outside Switzerland, and 4086 (9.4%) had at least a missing variable, leading to 25,260 (58.2%) PAH that were included in the analysis. For COPD, data from 148,504 PAH were initially assessed, of which 3361 (2.3%) were patients coming from outside Switzerland, and 10,074 (6.8%) had at least one missing variable, leading to 135,069 (90.3%) PAH included in the analysis.

The main characteristics of the PAH for asthma and COPD are summarized in Table 1. For asthma, women represented two-thirds of PAH, while age groups were evenly distributed within the PAH; non-Swiss nationals living in Switzerland represented one-quarter of PAH, and the Leman region accounted also for one-quarter of PAH. Most PAH for asthma had compulsory insurance, and three-quarters were admitted in an emergency setting; in one-third of the cases, the admission was due to the patient and not a doctor; almost 95% of PAH had a Charlson index below 4 (Table 1).

For COPD, men represented most of the cases, and over half of PAH were related to patients aged 70 years or older; Swiss nationals represented the majority of PAH and there was no evidence of a regional predominance. Almost all PAH for COPD had compulsory insurance, and two-thirds were admitted in an emergency setting; in one-fifth of the cases, the admission was due to the patient and not a doctor, and 95% of the PAH had a Charlson index of 2 or more (Table 1).

### 3.2. Trends of Potentially Avoidable Hospitalizations

The number and the EU-standardized rate (per 100,000 people) of PAH for asthma and COPD are summarized in Table 2 and in Figure 1 and Figure 2. For asthma, the number and rate increased from 1998 to 2001 and decreased afterwards, followed by a slight and continuous increase from 2008 onwards (Table 2 and Figure 1). For COPD, the number and rate increased continuously, with a slight attenuation during 2004–2010 (Table 2 and Figure 2).

### 3.3. Consequences of Potentially Avoidable Hospitalizations

The number of days for hospital stays (according to the OECD definition) and the number of beds exclusively dedicated to PAH for asthma and COPD are summarized in Table 3 and Figure 3 and Figure 4. For both conditions, the trends mimicked those of the rates, except that for asthma, where no increase was noted after 2008. In 2018, PAH for asthma would occupy 22 hospital beds during the whole year, this number increasing to 233 beds for COPD. Finally, in 2018, the estimated total costs of PAH amounted to 7.7 million CHF for asthma and 91.2 million CHF for COPD.

## 4. Discussion

According to recent data available from the OECD, Switzerland has a relatively low rate of potentially avoidable hospitalization for both asthma and COPD [25]. Nevertheless, the results from our study show that the rates of PAH due to asthma or COPD in adults aged 20 years or older have been increasing during the last two decades, notably for COPD. This concerning trend for PAH from COPD opposes Switzerland to the majority of the OECD countries, where decreasing COPD admission rates have been reported in recent years (2009–2019) [25]. Our results also show that the consequences of PAH for the Swiss health system, as in number of hospital days or costs, are far from trivial. These results might reflect a progressive reduction in accessibility and continuity in primary care, which has an impact on the timely identification and management of two common chronic respiratory diseases.

### 4.1. Characteristics of Potentially Avoidable Hospitalizations

In this study, two-thirds of PAH from asthma occurred in women. This finding is coherent with existing data regarding asthma, in which adult women have an increased prevalence and severity of the disease [26]. Surprisingly, the regional rate of PAH for asthma was high for the Leman area, corresponding to 24% of PAH, despite having only 15% of the Swiss population [27]. Considering COPD, and such as in the disease itself, older men represented most of the cases of PAH from COPD [28]. No regional predominance was identified as was found for PAH for asthma, which might suggest that individual patient factors (such as anxiety provoked by disease flares) and others related to disease management plans might also play a concurring role in the underlying mechanisms of PAH.

In our study, we have not found reasonable explanations for the regional disparities that can be a consequence of multiple and independent reasons. Further research might shed light on the impact of Swiss medical demographics (especially in primary care) and on the density of healthcare infrastructures (such as public hospitals or private clinics) in PAH. Switzerland has one of the highest proportions of foreign population residing in the country, so cultural aspects related to healthcare literacy and patterns of healthcare use can also be interesting as complimentary areas of investigation. In addition, recent evidence found the existence of socioeconomic gradients in PAH between 2014 and 2017 in Switzerland, as well as an important geographic variation between cantons [29].

The majority of the admissions of PAH occurred after an emergency visit (78.3% for asthma and 73.7% for COPD). Regarding asthma exacerbations, these can be adequately and timely treated by the patients themselves, and eventually in primary care if required [1]. It is likely that patients suffering from a respiratory exacerbation that is not responding to rescue treatment consider admission to the emergency unit where they can easily access treatment and thus be reassured by regular healthcare surveillance. Primary care providers need to reinforce access, even in out-of-hours, and provide specific and locally adapted recommendations for chronic respiratory patients to avoid, as much as possible, inadequate use of emergency care that can lead to subsequent PAH. Healthcare providers in the emergency units might require additional surveillance and safety networking for comorbid patients and, therefore, choose to hospitalize patients for observation, being less respectful of formal hospitalization criteria. On the other hand, it is also expected, from a public health perspective, that primary care providers provide respiratory patients with an explicit ambulatory pathway for the treatment of acute exacerbations. All those considerations demonstrate considerable room for improvement for the management of acute asthma and COPD exacerbations in the ambulatory setting.

In the analysis in this study, over two-thirds of COPD PAH patients had a Charlson index of ≥2. These numbers are aligned with the results of previous studies showing the presence of comorbidities as a particularly key factor influencing hospitalizations to ambulatory-care-sensitive conditions, especially in COPD [30,31,32]. Comorbidity in older patients can also lead to a more complex and difficult management of multiple chronic diseases in primary care, which might potentially reduce time allocated for the treatment of respiratory diseases.

### 4.2. Trends in Potentially Avoidable Hospitalizations

The EU-standardized rate of PAH for asthma increased from below 19/100,000 to almost 28/100,000 in 2000, declining afterwards and reversing the trend into a slightly increasing curve between 2008 and 2018. On the other hand, the EU-standardized rates of PAH for COPD almost doubled from 77.4/100,000 in 1998 to 142.7/100,000 in 2018. Seasonal fluctuations related to causal factors, such as viral infections, might explain changes in trends (2002, 2004, and 2014 for asthma; 2003 to 2006 for COPD), although influenza official statistics are not quite suggestive of it [33].

### 4.3. Implications of Potentially Avoidable Hospitalizations for Public Health

Hospitalizations from asthma or COPD can be a consequence of suboptimal disease management due to several preventable reasons, such as a lack of access to healthcare, insufficient or suboptimal pharmacological treatment, and inadequate disease knowledge, among many others. Even though accessible and continuous primary care services might play a role in preventing hospitalizations due to asthma and COPD, some studies have shown that this association might not be as linear as supposed [34]. In fact, individual factors, such as patients’ gender or age, as well as the presence of comorbidities, increase the risk of PAH independently of the quality of the primary care provided.

Healthcare systems are actually facing different challenges, such as the increase in non-communicable diseases and health inequities, as well as an ageing world population. If PAH are not adequately monitored and become neglected as a public health issue, the number of hospital stays, and corresponding number of occupied beds, will potentially increase in the upcoming years. In 2018, the number of hospital stays of PAH for asthma and COPD corresponded to the occupation of 255 beds for a full year. This is slightly below the median capacity of a tertiary care hospital in Switzerland (298 beds) that would be fully dedicated to managing PAH from two common chronic respiratory diseases [35]. Moreover, the combined costs of PAH for asthma and COPD for the year 2018 reached almost 100 million CHF, representing over half of the Swiss expenditure on prevention and health promotion (193 million CHF) [36]. Overall, our results suggest that PAH for asthma and COPD demand a considerable amount of health and economic resources. Thus, to effectively tackle PAH related to asthma and COPD, multidisciplinary public health strategies are urgently required to promote continuous and effective ambulatory management of chronic respiratory diseases and to increase access of vulnerable, comorbid, and older patients to primary care services [29].

### 4.4. Strengths and Limitations

Our study presents some relevant strengths. First, the use of exhaustive data with more than 21 million hospitalizations from all the Swiss regions increases the reliability of the obtained results. Second, the use of a long study period, such as 20 years, has permitted to compute trends and provide visual interpretation of the evolution of PAH through two decades. Third, the use of international recognized criteria for PAH allows valuable comparisons with the published literature, especially with existing findings from previous research that has been published in recent years with Swiss data [15,16,17]. This study also allowed an estimation of the costs of PAH for asthma and COPD, which, to our knowledge, was obtained for the first time in Switzerland for such an extended period of time.

There are, however, some limitations in our study. First, the quality and accuracy of hospital data used can be quite heterogeneous among Swiss cantons and regions due to different diagnostic coding habits from healthcare professionals. This lack of consistency might consequently diminish the internal validity of our results. However, this is a well-known limitation of research that is conducted by the use of aggregated data that depend on codification of inpatient main and secondary diagnosis. As stated by Eggli et al., the inpatient main diagnosis provides only a partial view of the cause of hospitalization and, hence, several adjustments, such as individual factors or morbidities, should be added to the indicators for ambulatory-care-sensitive conditions as a way to improve data analysis and interpretation [15]. Second, despite the regular use of PAH as internationally standardized indicators for interpreting the quality of primary care, elevated rates of ambulatory-care-sensitive conditions might be associated with heterogeneous approaches for hospital admission criteria and locally accepted procedures and finally not related to the ambulatory care itself [37]. This is one of the limitations of using PAH as a quality indicator that has been driving research in this field, aiming to improve its pertinence and clinical relevance [38]. Third, no adjustments were made to the regional healthcare supply, whether in terms of hospital care or primary care physicians, or to socioeconomic indicators, or even disease prevalence. This information can provide relevant clues and eventually shed light on some of the regional differences for accessibility and continuity of care. Finally, these findings apply to the Swiss health system and cannot be generalizable to other settings. Still, they provide important information for international comparisons and can reinforce the link between primary care and public health for better healthcare planning and improvement in the quality of care in Switzerland.

## 5. Conclusions

PAH from asthma and COPD in Switzerland have increased between 1998 and 2018, especially for COPD in the last decade, and represent a significant avoidable burden and cost for the Swiss healthcare system. Further research is needed to understand the reasons and driving forces behind PAH of these common respiratory diseases to be able to tackle this issue on a regional and national level with the active involvement of patients, healthcare providers, and stakeholders.

## Figures and Tables

**Figure 1 healthcare-11-01229-f001:**
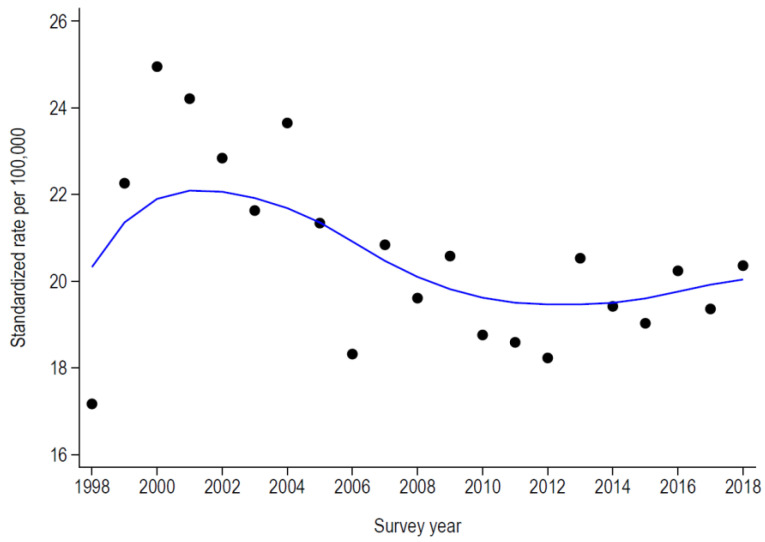
Evolution of the standardized rates of potentially avoidable hospitalizations for asthma in Switzerland, 1998–2018.

**Figure 2 healthcare-11-01229-f002:**
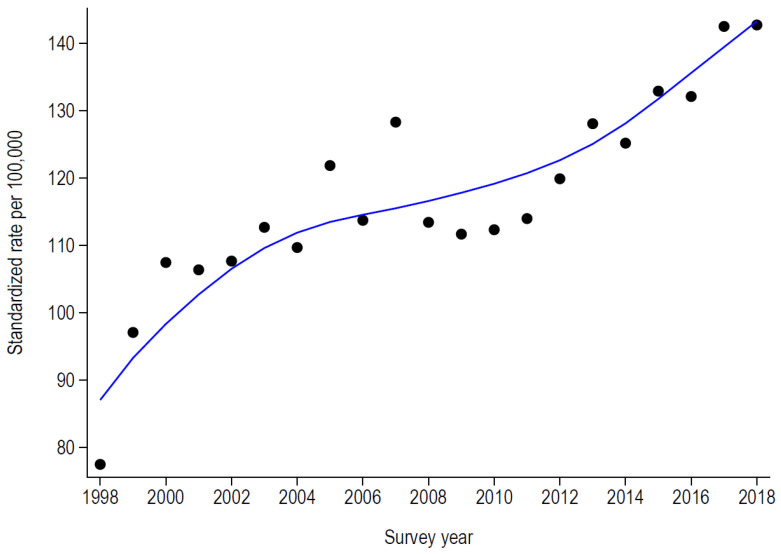
Evolution of the standardized rates of potentially avoidable hospitalizations for chronic obstructive pulmonary disease (COPD) in Switzerland, 1998–2018.

**Figure 3 healthcare-11-01229-f003:**
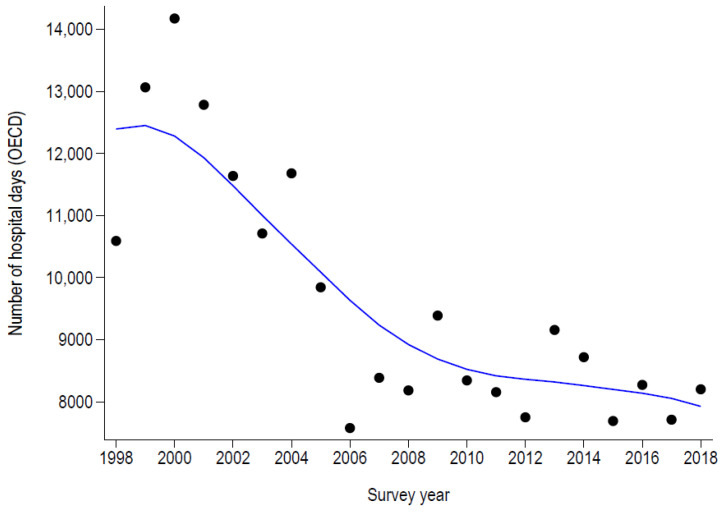
Number of days of potentially avoidable hospitalizations for asthma in Switzerland, 1998–2018.

**Figure 4 healthcare-11-01229-f004:**
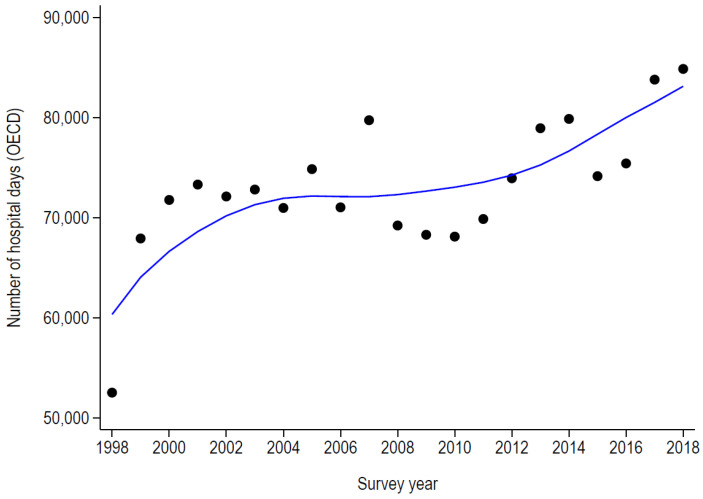
Number of days of potentially avoidable hospitalizations for chronic obstructive pulmonary disease in Switzerland, 1998–2018.

**Table 1 healthcare-11-01229-t001:** Main characteristics of potentially avoidable hospitalizations for asthma and chronic obstructive pulmonary disease (COPD) in Switzerland, 1998–2018.

Characteristics	Asthma	COPD
N	25,260	135,069
Women (%)	16,694 (66.1)	57,341 (42.5)
Age groups (%)		
[20, 30]	2720 (10.8)	454 (0.3)
[30, 40]	3146 (12.5)	1115 (0.8)
[40, 50]	3894 (15.4)	4254 (3.2)
[50, 60]	3912 (15.5)	15,527 (11.5)
[60, 70]	3671 (14.5)	33,956 (25.1)
[70, 80]	4247 (16.8)	47,337 (35.1)
[80, 90]	3140 (12.4)	29,011 (21.5)
[90, ∞]	530 (2.1)	3415 (2.5)
Swiss nationality (%)	19,076 (75.5)	116,881 (86.5)
Region (%)		
Leman	6015 (23.8)	24,824 (18.4)
Mittelland	4915 (19.5)	26,185 (19.4)
Northwest	4440 (17.6)	25,285 (18.7)
Zurich	3435 (13.6)	22,192 (16.4)
Eastern	3110 (12.3)	17,170 (12.7)
Central	1702 (6.7)	9795 (7.3)
Ticino	1643 (6.5)	9618 (7.1)
Compulsory insurance (%)	24,148 (95.6)	132,284 (97.9)
Emergency admission (%)	19,775 (78.3)	99,586 (73.7)
Patient’s decision to access healthcare (%)	9537 (37.8)	31,682 (23.5)
Charlson’s index categories (%)		
0–1	18,487 (73.2)	5750 (4.3)
2–3	5729 (22.7)	95,240 (70.5)
≥4	1044 (4.1)	34,079 (25.2)

**Table 2 healthcare-11-01229-t002:** Number and EU-standardized rate (per 100,000 people) of potentially avoidable hospitalizations for asthma and chronic obstructive pulmonary disease (COPD) in Switzerland, 1998–2018.

	Asthma		COPD	
Year	Number	Rate	Number	Rate
1998	864	17.17	3501	77.4
1999	1144	22.26	4453	97.0
2000	1285	24.95	4986	107.4
2001	1282	24.21	5048	106.3
2002	1206	22.84	5185	107.6
2003	1183	21.63	5514	112.6
2004	1309	23.65	5469	109.6
2005	1186	21.34	6188	121.8
2006	1037	18.32	5880	113.7
2007	1206	20.84	6758	128.3
2008	1148	19.61	6114	113.4
2009	1216	20.58	6135	111.6
2010	1126	18.76	6284	112.3
2011	1145	18.59	6483	113.9
2012	1141	18.23	6963	119.9
2013	1291	20.53	7599	128.0
2014	1244	19.42	7579	125.1
2015	1240	19.03	8197	132.9
2016	1340	20.24	8306	132.1
2017	1292	19.36	9114	142.5
2018	1375	20.36	9313	142.7

**Table 3 healthcare-11-01229-t003:** Number of days and beds occupied for potentially avoidable hospitalizations for asthma and chronic obstructive pulmonary disease (COPD) in Switzerland, 1998–2018.

	Asthma		COPD	
Year	Days	Occupied Beds	Days	Occupied Beds
1998	10,590	29	52,529	144
1999	13,064	36	67,929	186
2000	14,173	39	71,776	197
2001	12,783	35	73,314	201
2002	11,637	32	72,122	198
2003	10,710	29	72,815	199
2004	11,679	32	70,976	194
2005	9843	27	74,855	205
2006	7576	21	71,035	195
2007	8385	23	79,741	218
2008	8182	22	69,223	190
2009	9387	26	68,297	187
2010	8343	23	68,116	187
2011	8154	22	69,871	191
2012	7749	21	73,942	203
2013	9156	25	78,940	216
2014	8717	24	79,872	219
2015	7689	21	74,155	203
2016	8271	23	75,425	207
2017	7710	21	83,794	230
2018	8200	22	84,863	233

## Data Availability

The existing datasets used during the current study are not publicly available and can be provided upon justified request. Data are stored in the University Hospital of Lausanne under specific criteria. A copy of the contract between the Federal Office of Statistics and the University Hospital of Lausanne regarding data provision, security, and use can be provided upon justified request.

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
