# Peer review of "Potentially Avoidable Hospitalizations by Asthma and COPD in Switzerland from 1998 to 2018: A Cross-Sectional Study"

_healthcare, 2023, doi:10.3390/healthcare11091229_

Round 1

Reviewer 1 Report

The report from Alexandre Gouveia et al. investigated potentially avoidable hospitalizations (PAH) by asthma and COPD in Switzerland between 1998 and 2018. The findings are exhaustive and the quantitative parameters are adequately chosen following established guidelines. The topic is relevant to contemporary healthcare challenges. Although the scope and the impact of the study are limited to the analysis of one country (Switzerland), the experimental design and the analysis of the results are potentially of interest to a greater readership. I have only minor comments that could be addressed to improve the content:

1.       Lines 54-55: the formulation is inadequate. It should be rephrased with quantitative/comparative (socio-)economic indicators instead.

2.       The length of stay is mentioned in the variables but only appears as a number of global hospital days per survey year in the figures. It would be interesting to analyze the average LOS per hospitalization in the case of PAH vs all others.

3.       How many public and private hospitals were included in the analysis?

4.       The quantitative data concerning the total diagnosis of COPD and asthma in Switzerland is required to properly address the question of the trend for PAH associated with the admissions.

5.       Do the authors have an explanation regarding the regional rate of PAH in the Leman area (lines 185-187)?

6.       I am puzzled by the remark lines 262-264. Does it mean that the data cannot be trusted? How did the authors evaluate the accuracy of the raw acquisitions?

7.       Concerning ethics: it seems that no authorization was required to consult all the databases from the hospitals. Did the authors declare/register their investigation to a certified board dealing with health data?

Reviewer 2 Report

Very interesting study

“Seasonal fluctuations related to causal factors, such as viral infections, might explain changes in trends (2002, 2004 and 2014 for asthma; 2003 to 2006 for COPD), although influenza official statistics are not quite suggestive of it [28]”

Good idea, except it catches only part of the problem, as it excludes other infections. However, if there is a late winter spike in infections followed by spike of hospitalization, then there is an issue whether indeed we face untreated chronic problems or whether everything was under control until the last infection. You can easily test two things:

-seasonal trends – if a person was hospitalised in the summer, it’s easier to claim that it should have been avoidable if only those chronic problems were treated properly. 

-do you have data on pneumonia hospitalization? I mean you can easily check whether those two phenomena are suspiciously well correlated. (if they are, then you should add caveat)

Limitations: 

While you mentioned all key issues correctly, I think that you should slightly change emphasis on data quality and compatibility. You already pointed that based on (Charlson's index) patients have a long list of comorbities and medical personnel has high amount of leeway in picking what’s the main problem. I know, Switzerland, so I’d expect people collecting the data more diligently than what I have seen for other countries. I’d at least consider re-arranging the limitations in to two paragraph 1) classification mess 2) by occasion some real differences are possible

279 (and 281) “Supplementary Table 1: OCDE criteria” No French, use English: OECD

Reviewer 3 Report

The authors proposed an analysis of potentially avoidable hospitalizations caused by asthma and chronic obstructive bronchitis. The original article considers the characteristics of the patients and analyzes the derived costs. All the aspects useful for describing the topic in a coherent, rigorous and exhaustive way have been carefully considered. I personally found this article interesting and enjoyable to read. I congratulate the authors and recommend the publication.

The authors have already described very well the strengths and weaknesses of the same study. The main selling point is that they were the first to address potentially avoidable hospitalizations for COPD and asthma in Switzerland. They did it with rigor and relying on reliable data, applying international definitions and criteria without errors. A curiosity emerges, a seasonal trend of hospitalizations was not drawn (which would be expected in diseases such as asthma and COPD) and it was not correlated with the Swiss administrative geography (in the various Swiss cantons there could be differences, such as a different number of patients for each general practitioner).
